# Latent classes of anthropometric growth in early childhood using uni- and multivariate approaches in a South African birth cohort

**Noëlle van Biljon** [1]*, **Marilyn T. Lake**[2], **Liz Goddard**[2], **Maresa Botha**[2], **Heather J. Zar**[2], **Francesca Little**[1]

1 Department of Statistical Sciences, University of Cape Town, South Africa, 2 Department of Paediatrics and Child Health, and South African-MRC unit on Child & Adolescent Health, University of Cape Town, South Africa

* noellevanbiljon@gmail.com

## Abstract

### Background

Defining growth patterns during childhood is key to identifying future health risk and vulnerable periods for potential interventions. The aim of this study was to identify growth profiles in children from birth to five years in a South African birth cohort, the Drakenstein Child Health Study (DCHS) using a Latent Class Mixed Modelling (LCMM) approach.

### Methods

LCMM was used to identify underlying latent profiles of growth for univariate responses of standardized height, standardized weight, standardized body mass index and standardized weight-for-length/height measurements and multivariate response of joint standardized height and standardized weight measurements from birth to five years for a sample of 1143 children from a South African birth cohort, the Drakenstein Child Health Study (DCHS). Allocations across latent growth classes were compared to better understand the differences and similarities across the classes identified given different composite measures of height and weight as input.

### Results

Four classes of growth within standardized height ($n_1$=516, $n_2$=112, $n_3$=187, $n_4$=321) and standardized weight ($n_1$=263, $n_2$=150, $n_3$=584, $n_4$=142), three latent growth classes within Body Mass Index (BMI) ($n_1$=481, $n_2$=485, $n_3$=149) and Weight for length/height (WFH) ($n_1$=321, $n_2$=710, $n_3$=84) and five latent growth classes within the multivariate response of standardized height and standardized weight ($n_1$=318, $n_2$=205, $n_3$=75, $n_4$=296, $n_5$=242) were identified, each with distinct trajectories over childhood. A strong association (much greater or lesser than expected proportions (an increase by 25% in some cases), when compared to the proportion of abnormal growth features across the entire cohort) was found between various growth classes and abnormal growth features such as rapid weight gain, stunting, underweight and overweight.

**Data availability statement:** Researchers who are interested in datasets or collaborations can contact the PI, Heather Zar (heather.zar@uct.ac.za) or Data Manager Lesley Workman (lesley.workman@uct.ac.za) with a concept note outlining the request. More information can be found on our website (http://www.paediatrics.uct.ac.za/scah/dclhs).

**Funding:** This study was funded by Bill and Melinda Gates Foundation (OPP 1017641; OPP1017579), the NIH, USA (U54HG009824, U01AI110466], South African Medical Research Council, and the University of Cape Town (UCT). NVB was supported through a fellowship to UCT from the Developing Emerging African Leaders (DEAL) programme by The Carnegie Corporation of New York and a Departmental Fellowship from the Department of Statistical Sciences, UCT. The funders had no role in study design, data collection and analysis, decision to publish, or preparation of the manuscript.

**Competing interests:** The authors have declared that no competing interests exist.

## Conclusions

With the identification of these classes, a better understanding of distinct childhood growth trajectories and their predictors may be gained, informing interventions to promote optimal childhood growth.

## Introduction

A double burden of childhood malnutrition, undernutrition and obesity, is emerging in low- and middle-income countries (LMICs). In 2020, 23% of Southern African children under five years were stunted, 3% were wasted and 12% were overweight [1]. Despite this high burden of malnutrition, data on early growth trajectories are limited in LMICs.

Growth patterns established early in life may have long term consequences for health. Stunting, an indicator of chronic malnutrition [2,3] has been associated with infections in childhood and obesity in adulthood [4,5]. Childhood obesity is associated with development of several non-communicable diseases including cardiometabolic impairment, diabetes, and asthma [6,7]. Low birth weight, prevalent in LMICs, is associated with an increased risk of negative health consequences throughout childhood [8] and rapid catch-up growth from birth and 2 years may be associated with an increased risk of obesity.

Thus, defining growth patterns during childhood is key to identifying future health risk and vulnerable periods for potential interventions. The aim of this study was to identify growth profiles in children from birth to five years in a South African birth cohort, the Drakenstein Child Health Study (DCHS) using a Latent Class Mixed Modelling (LCMM) approach [9].

Additionally, the use of multivariate LCMM to identify latent growth classes based on joint height and weight measurements is proposed as this approach will not impose a restrictive, predefined relationship between height and weight as the use of zBMI or zWFH would when identifying distinct growth trajectories. No publications have identified latent classes using a multivariate response of height and weight or the comparison of latent growth class allocation across various growth responses.

## Methods

Growth profiles from birth through 5 years were derived for children in the DCHS a South African birth cohort. Pregnant women in Paarl, a poor peri-urban area in South Africa, were enrolled during their second trimester of pregnancy, from March 2012 - March 2015. Mothers were recruited at an antenatal clinic, and hence pregnancy was already established by the primary healthcare providers. Mothers were recruited at an antenatal clinic, and all who were over the age of 18 and intended to reside in the Paarl area for at least one year were eligible for this study. Gestational age was calculated using second trimester ultrasound. If this was unavailable, last menstrual period was used to calculate gestational age at the time of enrolment. The most common mode of delivery was natural birth, while 20.12% of the children within the study were delivered by caesarean-section. Mothers provided informed consent at enrolment and were re-consented annually; mother-child pairs were followed from birth through childhood as previously described [10]. Comprehensive growth measurements were taken by trained study staff at birth 6, 10, 14 weeks, 6, 9 and 12 months and then 6 monthly until 5 years (Fig 1 and S1 Fig). The most rapid period of growth through childhood is during infancy, specifically throughout the first 2 years, hence measurements were captured more frequently through this period.

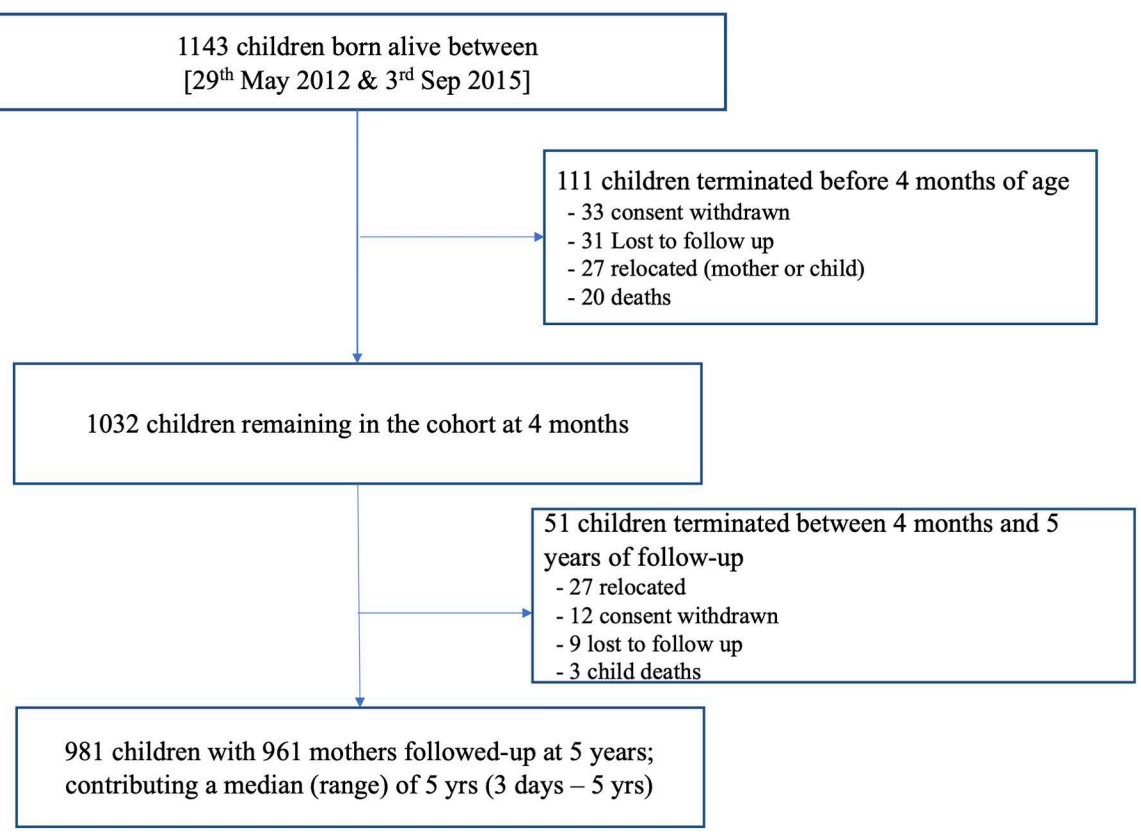

**Fig 1. The DCHS Cohort of children from birth until 5 years of age.**

This paper focuses on a model-based approach to identify latent growth profiles for standardized height and weight measurements, as well as the composite measures of these responses, namely standardised Body Mass Index (zBMI), standardised weight-for-height (zWFH) and a multivariate response of zHeight and zWeight (zHeight + zWeight).

## Ethics approval

The study was approved by the faculty of Health Sciences, Human Research Ethics Committee, University of Cape Town (401/2009), Stellenbosch University (N12/02/0002) and the Western Cape Provincial Health Research committee (2011RP45).

## Outcomes

Growth measurements included height and weight from which standardised body mass index (zBMI) and standardised weight for height (zWFH) were calculated. Height (recumbent length (recorded in centimetres) measured as distance from crown to foot using a Seca length-measuring mat (Seca; Hamburg, Germany) from birth until 18 months, after which this was recorded as standing height using a wall-mounted stadiometer (Panamed; Philippines)) and weight (the mass of a subject (to the nearest 10 g) in light or no clothing using a Tanita digital platform scale (TAN1584; IL, USA)) were measured by trained study staff. Each measurement was taken twice per visit per child to serve as technical replicates. Measures within 0.5 cm and 0.1 kg of each other for height and weight respectively were acceptable and the first measurement was used. If measures were not within this acceptable range, a third measurement was

taken and used. Anonymized anthropometric data used in this analysis were first accessed on the 13th of June 2022. For comparability, standardised growth responses (with respect to Growth References) were analysed within this report; these are denoted using "zGrowthResponse" such as zHeight for standardised height and zWeight for standardised weight. Growth responses as well as the number of observations per visit are summarised in S1 Table.

Fenton growth references were used to calculate standardised growth scores for prematurely born children (before 37 weeks' gestation); at birth and until 50 weeks postmenstrual age. WHO references were then used for both preterm and full term infants; where WHO was used for preterm infants aged between 50 weeks postmenstrual age onwards (with gestational age-corrected measurements up until 2 years), and WHO used for all measurements taken from full term infants at birth until 5 years [11–13]. Standardised height (zHeight; derived using WHO and Fenton reference ranges), standardised weight (zWeight; derived using WHO and Fenton reference ranges), standardised body mass index (zBMI; weight in kg divided by height in meters squared, standardised using WHO reference ranges) and standardised weight for length/ height (zWFH; derived using WHO reference ranges) were calculated.

To identify children that experienced rapid weight gain, a change in zWeight equal or greater than 0.67 within the first nine months of life was used, as per convention, although there is disagreement on the optimal time frame to use [14].

## Statistical analysis

The modelling of the respective growth measures involved two overarching approaches: 1) Univariate- and 2) Multivariate Latent Class Mixed Models. Within the univariate modelling section each growth measure was described and analysed independently. The responses considered with this approach were zHeight, zWeight, zBMI and zWFH. The multivariate approach allowed the growth measures to be analysed together, thus identifying groups of subjects that follow similar trajectories while considering more than one growth response. Using this approach, latent classes from a model based on zHeight and zWeight was considered as an alternative to latent classes from the individual models for the calculated composite scores, zBMI and zWFH.

Prior to LCMM, univariate mixed effect models for the mean response were fit to each individual growth measure to determine the optimal structure for modelling the association with time. Both linear and non-linear relationships between time and the growth measurements were considered to ensure the best model fit, including linear, cubic splines, fractional polynomials, and piecewise linear splines. For growth measurement these four different model formulations with differing knot choices for the spline options were compared to each other. The piece-wise linear spline formulation was chosen for inclusion in the latent trajectory modelling based on model fit and ease of interpretation. Thus, for each growth measurement, a piece-wise linear spline was used to describe the change in the measure with increasing age. Between 3 and 4 knots were placed between ages 0.25 and 2.5 years for the different growth measures. The different model formulations and the location and number of knots were chosen to minimise AIC and thus allow for the best fit possible. Details of the final form of the mixed effect model for each of the responses are found within the supplementary materials – Part 2: Extended Statistical Methods.

Latent classes were identified using the latent class mixed model (LCMM) approach [9] without adjusting for any covariates. The choice of the number of latent classes was accomplished through the use of various fit statistics. For each growth measure, the latent class mixed effect model was fit with class numbers ranging from 1 to 6. The number of classes resulting in the lowest BIC (a measure of model fit to the data), largest Entropy (a measure of

the level of discrimination between the classes), and lowest Integrated Classification Likelihood (ICL, a measure of fit conditional on class discrimination) were selected subject to their size being greater than 5% of the sample and their stability.

The stability of the identified classes and profiles thereof were validated through internal cross-validation. This validation step involved randomly selecting approximately 50% of the subjects and refitting the latent class mixed effect model with the chosen number of classes. This was repeated with 10 different random samples. The response profile of the classes identified were plotted together on one set of axes and the class number that appeared to have the most consistency with respect to the class trajectories was taken into consideration in addition to the fit statistics.

LCMM can be extended to model multiple responses simultaneously by fitting multivariate mixed models and multivariate latent class mixed models for multivariate longitudinal response variables. This approach was used to identify latent growth classes within the combined responses of zHeight and zWeight, defining each longitudinal dimension as a latent process (S2 Fig), to allow for more flexible relationships between zHeight and zWeight within the multivariate setting [9].

## Results

Amongst 1137 mothers enrolled there were 1143 children live born, with 981 (85.8%) followed through 5 years of age with detailed measurements as shown in Fig 1. Twenty-three (2%) children died in infancy and 139 (12.2%) children were lost to follow up, the majority of which occurred before 6 months of age. All growth measurements were recorded at each visit, however some measurements were excluded due to instrumental or measurement error.

### Study cohort

Six-hundred and ninety-five (60.8%) of the mothers did not complete secondary schooling education, while 988 (86.4%) were members of households where the average monthly income was below R5000 (approximately USD276). One hundred and ninety-one (16.7%) children were born prematurely, the majority late preterm (born between 34-37 weeks gestational age). There were 247 (21.6%) HIV-exposed but uninfected children, with only 2 HIV-infected. Characteristics of included children were similar to those excluded; S2 Table summarises the total, included and excluded cohort with respect to maternal, child and socio-economic characteristics.

**Average trajectory of growth responses.** Fig 2 illustrates the standardised growth responses from birth until five years. As these observations represent standardized growth scores, the average trajectory should follow the line of $y = 0$. On average, lower zHeight and zWeight scores from what were anticipated according to WHO references were observed. Larger than expected zBMI and zWFH scores were observed, indicating that while children were below expected weight, their height deviated even further below expectation, resulting in above expected zBMI and zWFH. The trajectories of the unstandardised, observed, growth responses are illustrated within S3 Fig and S1 Table in the supplementary materials.

### Describing growth responses over time

The mean profiles for the zHeight and zWeight, zBMI and zWFH standardized measurements over time as estimated by the piecewise linear spline models are illustrated in the supplementary materials (S3 Table and S4 Fig). The early placement of the knots with no knots beyond age 2.5 reflects that most of the changes in trajectory structure occurred before two years of age. These changes in trajectory structure may also be a feature of more frequent

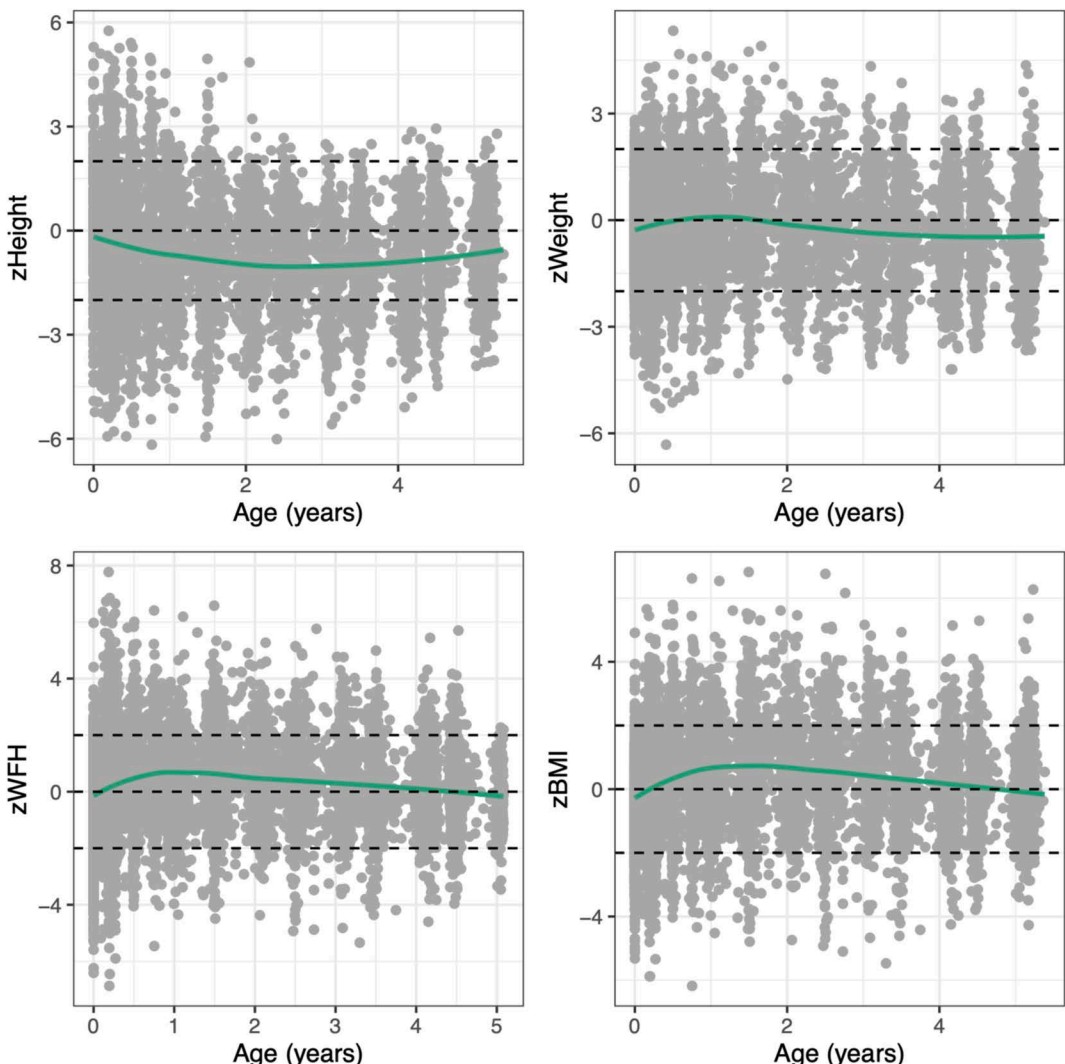

**Fig 2. Standardized growth measurements from Birth until age 5 with a smoothed average trajectory.**

data collection during the infancy. To determine whether the choice of knot location impacted the subsequently identified latent profiles, the longitudinal profiles given additional knots placed at later time points (ages 0.25,0.75,1,1.5,2,3,4; S5 Fig) were compared to those described in S3 Fig. The profiles did not show clear deviation, and thus the knot locations specified in S3 Table were used.

## Selecting K, the number of latent classes

The process whereby the number of latent classes (k) for the trajectories of zHeight, zWeight, zBMI, zWFH and zHeight + zWeight is illustrated within the supplementary materials – Part 3.3. For zHeight and zWeight, four latent classes were identified as optimal (S6 and S7 Figs). Three latent classes were identified as optimal within zBMI and zWFH (S8 and S9 Figs). When considering the multivariate zHeight + zWeight model, five latent classes were identified as optimal (S8 and S9 Figs).

## Growth trajectories

**Univariate trajectories.** Each response, described using the broken stick model previously indicated, fit with the latent class mixed model given the optimal number of k is illustrated in Fig 3, thus illustrating the trajectories of each latent growth class identified within the respective growth responses. The model estimates from the fixed effect components can be found within S4–S9 Tables in the Supplementary Materials.

**zHeight.** The four latent classes for zHeight are illustrated in Fig 3 (A). The following features were observed: Firstly, zHeight profiles below the expected are observed (this was also reflected in zWeight profiles). Secondly, much of the differences in structure of

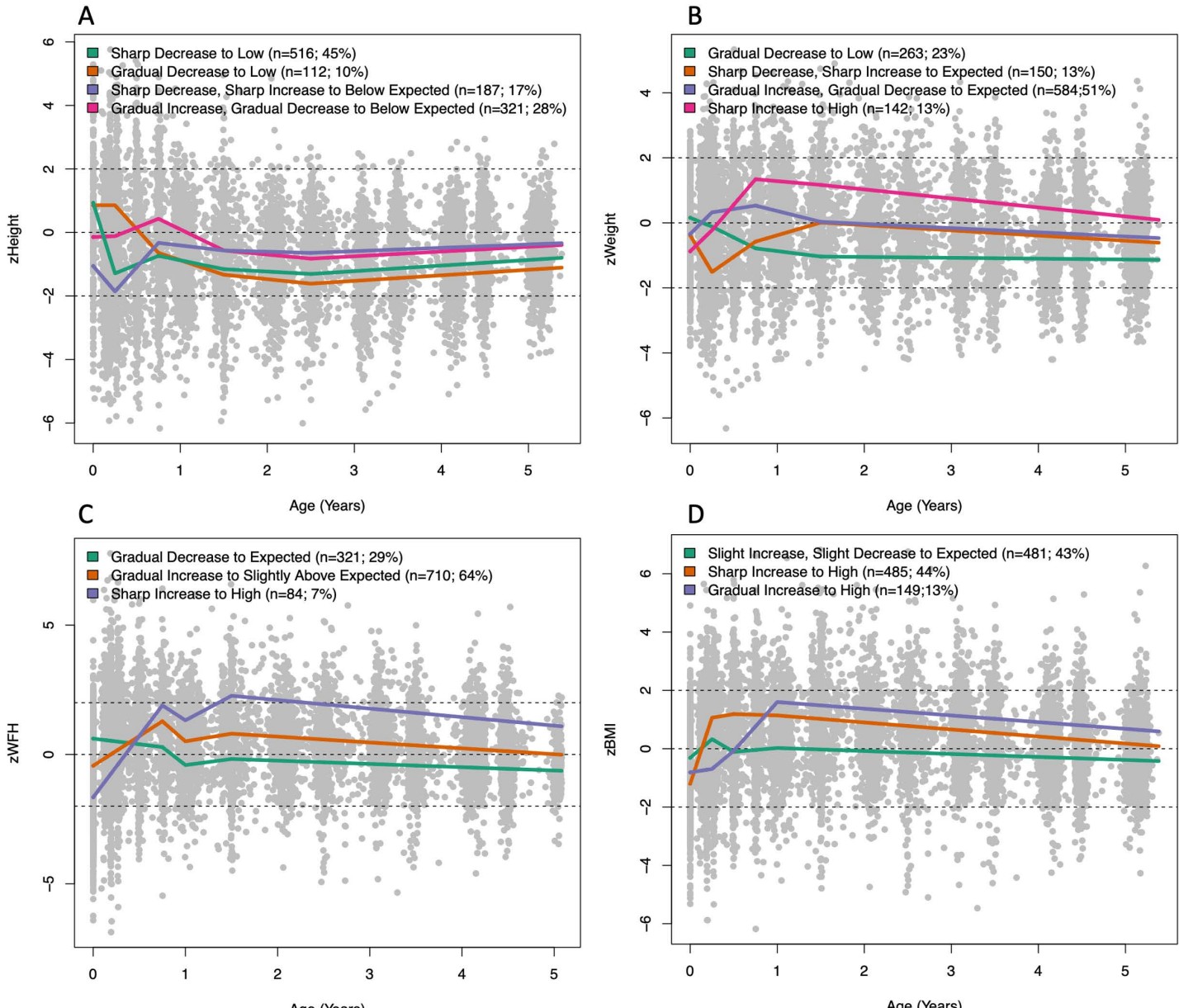

**Fig 3. The average trajectories of latent classes identified within A) zHeight, B) zWeight, C) zWFH and D) zBMI from birth until age of five.**

trajectories occur before 1.5 years, after which the four trajectories settle at different levels. This change in profiles prior to and after 1.5 years is reflected in the early changes in slope and later levels naming of the four identified classes (for example, sharp increase to expected levels). Two classes that settled slightly closer to the expected height are seen, one class of 321 (labelled "Gradual Increase, Gradual Decrease to Below Expected") children who started at expected height, increased to slightly higher heights than expected and then settled at a level of less than one standard deviation below the expected level. Another class (labelled "Sharp Increase, Sharp Decrease to Below Expected") includes 187 children who started with lower heights than expected but by the age of 1 year have caught up to a level less than one standard deviation below the expected height. A further two classes show children who were taller than expected at birth, but who then attained a level up to two standard deviations below the expected height, one group of 516 children (labelled "Sharp Decrease to Low") earlier than the other group of 112 children (labelled "Gradual Decrease to Low").

**zWeight.**  Four latent classes were identified within zWeight. Similar to zHeight, the heterogeneity in structure is seen in the profiles prior to 1.5 years. Two classes (labelled "Gradual Increase, Gradual Decrease to Expected" and "Sharp Decrease, Sharp Increase to Expected") initialize close to the expected zWeight at birth and diverge within the first 1.5 years, one class slightly above expected zWeight and another dipping well below expected, after which they meet and show subjects that follow expected weight for age from 1.5 to 5 years old (Fig 3 (B)). Two classes with birthweights as expected who then deviated from expected weight profiles were observed, resulting in a class with above expected weights (labelled "Sharp Increase to High") and a class with lower-than-expected weights (labelled "Gradual Decrease to Low").

**zWFH.**  Three classes were identified for zWFH (Fig 3 (C)). A class that almost follows the expected weight-for-height from birth until five years (labelled "Gradual Decrease to Expected"), and two classes with consistently greater weights given their height and sex. The group with the greatest weight for height (labelled "Sharp Increase to High"), showed the smallest weight-for-height at birth.

**zBMI.**  Three classes were identified for zBMI, all of which were characterised by early increasing zBMI measurements (Fig 3 (D)). A class that closely follows the expected zBMI from birth until five years, except for a brief fluctuation above the expected BMI before 6 months of age (labelled "Sharp Increase, Sharp Decrease to Expected"). Two classes with smaller than expected zBMI at birth which both settle above the expected zBMI. One class settles earlier on, at 3 months of age (labelled "Sharp Increase to High") while the second settles around one year of age (labelled "Gradual Increase to High").

Fig 3 interestingly also show that those with the most extreme lower than expected measurements at or soon after birth, settled at the highest levels, and vice versa, indicating an over-correction of sorts.

**Multivariate trajectories.**  Fig 4 illustrates the zHeight and zWeight profiles for the five classes determined for the joint zHeight + zWeight measurements. Three hundred and eighteen children were allocated to a class with low zHeight and zWeight from birth until five years (labelled "i"). Both zHeight and zWeight was above expected at birth but slowly decreased to below expected at birth for 205 children (labelled "ii"). A small group of children (n = 75) showed a delay in both zHeight and zWeight in the first few months of growth, however this did not appear to have long lasting impacts on zHeight or zWeight trajectories (labelled "iii"). A class of 296 children illustrate weight and height measurements equal to what was expected prior to age 1 however in subsequent observations these children were shorter than expected given their age and sex, while

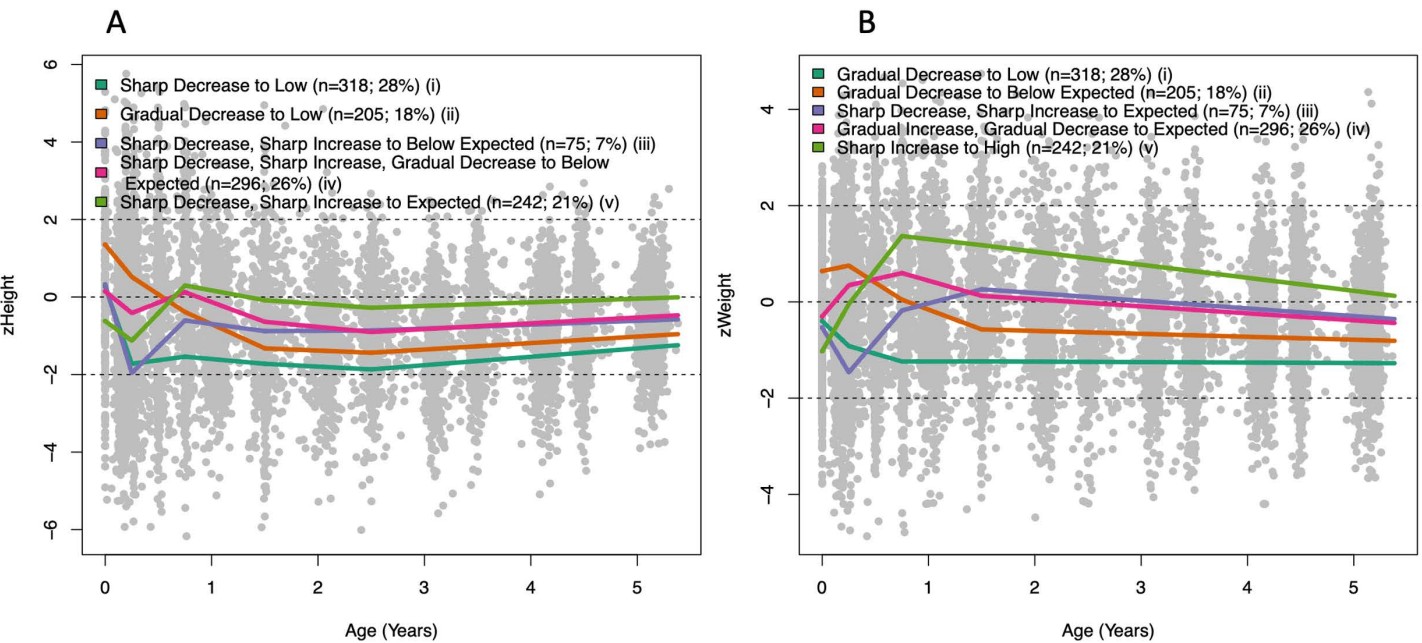

**Fig 4. The average trajectories of four classes identified within zHeight + zWeight from birth until age of five, illustrated using A) zHeight and B) zWeight.**

maintaining an expected weight (labelled "iv"). The last identified class shows individuals with a low zWeight at birth which steadily increased within their first year and remained above what was expected (labelled "v"). This same group was also shorter than expected at birth, with a slight delay in height growth during the first year of life, after which they followed expected height for age and sex.

The accordance or dissimilarity of zHeight and zWeight trajectories within the latent classes is of interest. Class (iv) shows different profiles in particular before 1.5 years where an increasing zWeight and decreasing zHeight can be seen. Additionally, class (v) also shows an increasing zWeight with decreasing zHeight within the first few months of life.

To determine whether the choice of knot location impacted the subsequently identified latent profiles, the latent class identification process was repeated with additional knots placed at later time points. The extended knot locations used were (0.25,0.75,1,1.5,2,3,4) years for all responses considered. The results from this longitudinal specification are presented in the supplementary materials (S12–S17 Figs). The results did not differ, and thus the knot locations specified in S3 Table were used.

## Comparison of latent class allocations

The correspondence between latent class allocations for individual children based on different growth measurements and on allocations based on univariate versus multivariate models are shown in Tables 1 and S10–S12. Table 1 illustrates the correspondence between allocations based on zHeight and zWeight individually and zHeight and zWeight within a joint model in a three-way contingency table. S10 and S11 Tables summarise the agreement between composite measures of height and weight (zBMI and zWFH) and the class allocations based on models for joint zWeight and zHeight. Finally, S12 Table summarises the agreement between the latent classes identified given zWFH or zBMI.

**Table 1. Tabulation of multivariate group allocation versus.**

| Multivariate zHeight: | Gradual Decrease to Low | Gradual Increase; Gradual Decrease to Below Expected | Sharp Decrease to Low | Sharp Decrease; Sharp Increase to Below Expected | All Height |
|---|---|---|---|---|---|
| (a) univariate height group allocation | | | | | |
| i: zHeight sharp decrease to low zWeight gradual decrease to low | 10(3.14%) | 48(15.09%) | 218(68.55%) | 42(13.21%) | 318(100%) |
| ii: zHeight gradual decrease to low zWeight gradual decrease to below expected | 84(41.0%) | 46(22.4%) | 75(36.6%) | 0(0%) | 205(100%) |
| iii: zHeight sharp decrease, sharp increase to below expected zWeight sharp decrease, sharp increase to expected | 3(4.0%) | 12(16.0%) | 42(56.0%) | 18(24.0%) | 75(100%) |
| iv: zHeight sharp decrease, sharp increase, gradual decrease to below expected zWeight gradual increase, gradual decrease to expected | 12(4.1%) | 138(46.6%) | 128(43.2%) | 18(6.1%) | 296(100%) |
| v: zHeight sharp decrease, sharp increase to expected zWeight sharp increase to high | 3(1.3%) | 77(31.8) | 53(21.9%) | 109(45%) | 242(100%) |
| TOTAL | 112(9.9%) | 321(28.3%) | 516(45.4%) | 187(16.5%) | 1136(100%) |
| (b) univariate weight group allocation | | | | | |

| Multivariate zWeight: | Gradual Decrease to Low | Gradual Increase; Gradual Decrease to Expected | Sharp Decrease; Sharp Increase to Expected | Sharp Increase to High | All Weight |
|---|---|---|---|---|---|
| i: zHeight sharp decrease to low zWeight gradual decrease to low | 138(43.4%) | 118(37.1%) | 61(19.2%) | 1(0.3%) | 318(100%) |
| ii: zHeight gradual decrease to low zWeight gradual decrease to below expected | 105(51.2%) | 100(48.8%) | 0 (0%) | 0 (0%) | 205(100%) |
| iii: zHeight sharp decrease, sharp increase to below expected zWeight sharp decrease, sharp increase to expected | 2(2.7%) | 16(21.3%) | 51(68%) | 6(8%) | 75(100%) |
| iv: zHeight sharp decrease, sharp increase, gradual decrease to below expected zWeight gradual increase, gradual decrease to expected | 18(6.1%) | 265(89.5%) | 5(1.7%) | 8(2.7%) | 296(100%) |
| v: zHeight sharp decrease, sharp increase to expected zWeight sharp increase to high | 0 (0%) | 82(33.9%) | 33(13.6%) | 127(52.5%) | 242(100%) |
| TOTAL | 263(23.2%) | 581(51.1%) | 150(13.2%) | 142(12.5%) | 1136(100%) |

## Comparison of univariate and multivariate zHeight and zWeight latent class allocations

Table 1(A) and (B) cross-tabulate the multivariate allocations to the univariate allocations of subjects based on height and weight, respectively. These tables show the number of children in each cross-tabulation and percentages of the multivariate classifications allocated to the different univariate height or weight classes. When cell frequencies exceed the distribution of the total cohort across the height or weight classes, they are indicative of a strong association with the specific multivariate class.

There is a strong agreement between identified trajectories such as within multivariate class (i) where the zHeight and zWeight classes with the strongest association describe the same sharp or gradual decrease to low trajectories as described by multivariate class (i).

Multivariate class (iii) showed identical sharp decrease, sharp increase to expected zWeight profiles when comparing the univariate and multivariate classes with the strongest associations, while the zHeight trajectories for these classes differed between the multivariate allocation that allowed for a sharp increase to below expected and univariate allocation that reflected a sharp decrease to low levels.

Similarly, multivariate Class (v) showed similar increasing trajectories to high levels for zWeight as identified by the univariate profiles, but slightly differing profiles for zHeight that settled at different levels for the multivariate and univariate profiles.

For multivariate class (ii), the converse of this was observed where the zHeight profiles descriptions based on multivariate and univariate allocations with the strongest association were identical, reflecting a gradual decrease to a low level, while the zWeight profiles differ with respect to the level at which the profiles settled.

Finally, multivariate class (iv) showed identical zWeight profiles across univariate and multivariate allocations with the strongest association, while the zHeight profiles differ both with respect to level and rate of change prior to 1.5 years.

In summary, it appears that based on the association across the univariate and multivariate class allocations, zWeight profiles appear to be in agreement more frequently than zHeight allocations. Additionally, the multivariate approach allowed for flexible profiles of one measurement (say zHeight) while holding the profiles for the other measurement (say zWeight) constant. For example, multivariate classes (i) and (ii) both reflect gradual decreases for zWeight but class(i) capture a sharp decrease in zHeight while class(ii) allowed for a more gradual decrease in zHeight. Class (ii) additionally also included a significant number of children whose weights first increased before showing a gradual decrease to their expected weight. None of these associations were perfect, making it difficult to get a clear assessment of the additional information conveyed by the multivariate approach.

## Comparison of class allocations based on composite measures (zWFH and zBMI) to that based on joint zHeight and zWeight measurements

Comparison of underlying latent classes identified by the joint zHeight and zWeight LCMM model and the latent classes identified for zWFH and zBMI, are shown in S10 and S11 Tables in the supplementary material. They illustrate a lack of strong agreement between latent profiles identified by the multivariate LCMM model and the individual zWFH and zBMI models, thereby emphasizing the flexibility of the multivariate LCMM approach to pick up heterogeneous behaviours in zWeight and zHeight that may not be evident in the zWFH and zBMI measures that have more restrictive associations between weight and height.

## Association with abnormal growth

To classify Rapid Weight Gainers (RWG), children were identified who experienced an increase in weight between birth and 9 months that was greater than 0.67 standardized units. The proportion of RWG children within each identified growth trajectory for weight is shown in Table 2.

Most children identified as RWG were allocated to the "Gradual Increase; Gradual Decrease to Expected" and "Sharp Increase to High" zWeight classes, while very few were allocated to the "Gradual Decrease to Low" and "Sharp Decrease, Sharp Increase to Expected" classes (Table 2). 93% of children allocated to the "Sharp Increase to High" class experienced RWG, showing an association with RWG and the high zWeight trajectory. Within the "Gradual Increase; Gradual Decrease to Expected" class, 56% experienced RWG. Thus, this class is almost equally represented by children who experienced RWG, and those who did not, who have settled at a normal weight.

**Table 2. Proportion of children that experienced rapid weight gain within the first 9 months of life within the four zWeight growth classes.**

|  | Gradual Decrease to Low | Sharp Decrease, Sharp Increase to Expected | Gradual Increase; Gradual Decrease to Expected | Sharp Increase to High |
|---|---|---|---|---|
| **Normal WG** | 235 (98.7%) | 108 (86.4%) | 206 (43.7%) | 9 (6.8%) |
| **RWG** | 3 (1.3%) | 17 (13.6%) | 265 (56.3%) | 124 (93.2%) |

The proportion of those that experienced RWG within other growth trajectories is illustrated within the supplementary materials (S13–S15 Tables). Within zBMI, both "Increase to High" trajectories comprised over 50% RWG individuals, this is also seen within the zWFH "Sharp Increase to High" trajectory. Otherwise, the zBMI and zWFH classes do not show a clear association with RWG, which is to be expected as zBMI and zWFH measures consider both zHeight and zWeight profiles. Within the multivariate zHeight + zWeight classes a large percentage of RWG within those allocated to the multivariate classes (iv) and (v) (61% and 81% respectively) occurred, in line with results seen in zWeight.

The number of children that ever experienced some form of abnormal growth, namely stunting (zHeight Score < -2), overweight (zWeight Score > 2) or underweight (zWeight Score < -2), at one or more time points between birth and 5 years were identified (Tables 3–4). Both "Decrease to Low" (Sharp and Gradual) classes, based on zHeight, show over 50% of individuals classified as stunted at one or more time points (Table 3). Within the "Gradual Increase, Gradual Decrease to Below Expected" class 34% of subjects were classified as stunted at one or more time points, and within the "Sharp Decrease, Sharp Increase to Below Expected" class the largest percentage of 73% of individuals classified as stunted at one or more time points. The largest percentage of children identified as underweight at one or more time points, 50%, were allocated to the "Sharp Decrease, Sharp Increase to Expected" class when considering zWeight (Table 4). Meanwhile, the largest percentage of children identified as overweight at one or more time points, 41%, were allocated to the "Sharp Increase to High" class when considering zWeight (Table 4). These identified latent profiles thus indicate children with abnormal growth features across the 5-year period rather than at isolated ages.

## Discussion

The structure and heterogeneity of growth profiles for children in a LMIC African birth cohort characterised using LCMM identified distinct growth classes from birth through 5 years. Through these latent classes children were identified who deviated from expected growth patterns and their progression through early childhood was tracked. We identified a clear association between univariate zWeight and zHeight classes and zHeight + zWeight classes and abnormal growth patterns of stunting, underweight, overweight and rapid weight gain.

**Table 3. Proportion of ever stunted children between birth and 5 years as defined by zHeight scores.**

|  | Sharp Decrease to Low | Gradual Decrease to Low | Sharp Decrease, Sharp Increase to Below Expected | Gradual Increase, Gradual Decrease to Below Expected | Entire Cohort |
|---|---|---|---|---|---|
| **Stunted** | 263 (51%) | 61 (55%) | 136 (73%) | 110 (34%) | 570 (50%) |
| **Total** | 516 | 112 | 187 | 321 | 1136 |

**Table 4. Proportion of children ever underweight or overweight between birth and 5 years as defined by zWeight scores.**

|  | Gradual Decrease to Low | Sharp Decrease, Sharp Increase to Expected | Gradual Increase; Gradual Decrease to Expected | Sharp Increase to High | Entire Cohort |
|---|---|---|---|---|---|
| **Underweight** | 91 (35%) | 75 (50%) | 76 (13%) | 44 (31%) | 286 (25%) |
| **Overweight** | 31 (12%) | 18 (12%) | 77 (13%) | 58 (41%) | 184 (16%) |
| **Total** | 263 | 150 | 584 | 142 | 1139 |

This is the first study to identify latent growth classes using a joint growth response made up of zHeight and zWeight during childhood. Additionally, the trajectories identified using this joint response were contrasted against those identified using zBMI, zWeight, zHeight and zWFH. The results were internally validated using two specifications of the longitudinal process, through the use of standardised and unstandardised growth responses as well as through repeating the LCMM using internal cross-validation using randomly sampled subsets. Consequently, there is confidence in the latent profiles identified within this report.

Growth responses may be analysed within their observed or standardised formats. The observed, unstandardised approach does not control for prematurity, age or gender. Thus, a key advantage of using standardised growth responses is the generalisability of trajectories, as the impact of gender or gestational age on growth has been removed from the process of latent class identification.

A Cluster Analysis approach could have been used to identify latent groups with respect to growth instead of the LCMM approach. This would have required the definition and analysis of features of a given growth profiles as opposed to the longitudinal growth measurements. Gough et al., used K-means clustering approaches to identify latent trajectories within standardised height (zHeight) scores and identified four classes within this growth response [15]. Similarly, Mebrahtu et al., identified three classes within standardised weight (zWeight) scores using the GMM approach [16]. Here, within zHeight and zWeight four classes were identified, while zHeight + zWeight lead to the identification of five latent profiles. The classes identified using the multivariate approach were not identical to those found using the univariate approach, and additionally allocation to these classes was not identical. This shows the benefit of considering both the uni- and multivariate approaches when identifying latent growth trajectories. The multivariate approach considers both the change of zHeight and zWeight responses over time that are adjusted for age and sex, with an unrestricted relationship unlike zBMI and zWFH. This also reiterates that the multivariate approach allows the identification of latent trajectories that consider the change of both height and weight independently over time. The univariate and multivariate approaches consistently identified a low zHeight class, which may serve as a stunted, or at high risk of stunting class, a low zWeight class (those at greater risk of underweight) and a high zWeight class (those at greater risk of overweight).

Three growth classes were identified within zBMI and zWFH scores between birth and five years. In contrast, Rickman et al. identified four latent classes within zWFH, however they considered zWFH from birth until two years within a Western Kenyan cohort. While they identified an additional class, there is strong agreement between the trajectories identified by Rickman et al. and those within this study; the difference being with an additional low zWFH class identified by Rickman et al. [17]. It is possible that cohort differences may explain the difference in profile numbers identified here as Rickman et al., observed an increased risk of suboptimal growth - considering 8 distinct growth measures - within HIV-exposed uninfected children.

Stein et al., (2010) compared the patterns of height in early childhood (birth to age 5) in five birth cohorts from distinct populations around the globe (Brazil, Guatemala, India, Philippines and South Africa) [18]. Considering the same cohorts, Poveda Rey et al., (2021) looked at weight throughout early childhood [19]. These studies illustrated diversity in growth profiles across multiple LMIC nations. The Brazilian cohort mostly followed expected standardised height and expected standardised weight within the first five years of life. While the Guatemalan, South African and Indian cohorts were born smaller (with respect to length) than the expected; the South African cohort recovered by the age of five while the Guatemalan and Indian cohorts did not. The Guatemalan, Filipino and South African cohorts were born weighing slightly less than expected. The South African cohort did not recover, nor did they

show further reduction in expected weight. However, the Guatemalan and Filipino cohorts did show further reduction in standardised weight scores, to a greater extent for the Filipino cohort. The Filipino cohort was born with an expected height but decelerated to much smaller than expected by one year of age. The Indian cohort was born with the smallest standardised weight which did not improve through early childhood. Interestingly, all cohorts showed a decreased standardised height at two years compared to birth; while some were able to recover from this deceleration (Brazil, South Africa), other cohorts did not (Guatemala, India, Philippines).

Previous studies that make use of latent class analysis to identify groups of children with similar growth trajectories primarily focus on BMI as an indicator of early onset obesity [20,21]. Robinson et al., reviewed all publications post 2000 that investigated latent class identification within BMI responses [21]. Of the eight studies found, six identified four clear trajectories within BMI during childhood [21]. Accordingly, identification of latent classes within BMI or zBMI is seen more frequently in the literature than zWFH. Most commonly, three or four latent classes are identified within zBMI or BMI, however these studies often focus on growth from the age of four onwards [21–24]. Wang et al., found dynamic BMI growth patterns (such as BMI catch-up and stable overweight) were more predictive, and thus more informative, than static BMI measurements of cardiovascular structure and function in early childhood [25]. Rapid weight gain in early childhood (or catch-up growth) has been associated with an increased risk of overweight, obesity and other chronic diseases during later childhood and adulthood [26–29]. Additionally, catch-up growth has been associated with various factors such as altered insulin metabolism [30,31] and hence an increased risk of Type II diabetes [32], increased systolic blood pressure [33], increased risk of coronary heart disease [34] and an increased risk of childhood asthma [35] Various factors may modify the risk of rapid weight gain or obesity, including bottle feeding, shorter gestation age and being firstborn [36,37].

Conventionally weight-for-height/length (zWFH) is used to assess over/under-weight status in children below the age of 2 years, while Body Mass Index (BMI) is used from 2 years onwards [38]. Weight-for-height and standardised BMI (zBMI) scores are calculated using WHO charts; both represent the relationship between height and weight. However, zBMI scores are standardised by age while zWFH scores are not [11]. Instead, these are created using charts that represent the expected relationship between height and weight between birth and two years or two years and five years of age [39]. Aris et al., found that the use of zBMI or zWFH before 2 years did not impact the ability to predict future adiposity or cardiometabolic outcomes in children, suggesting these scores may be equally beneficial.

A clear limitation for the use of zWFH is the lack of age-dependent standardisation, which is resolved when one makes use of zBMI [11,39]. However, a shortcoming of the use of BMI to study future risk of obesity, is the fact that BMI does not consistently reflect body composition [40]. It has even been suggested that BMI and BMI increase during early childhood is more predictive of lean mass than adiposity during adulthood [41]. Body composition, fat mass and fat-free mass may be measured, and subsequent latent classes have been identified within the observations of fat mass and fat-free mass independently [42]. However, further research is needed to identify whether this may be a better predictor of obesity in children. Additionally, this requires dedicated machinery that may not be available at many care facilities in Low to Middle Income (LMIC) countries. Here, the trajectories identified within zBMI and zWFH as well as allocation to these classes across these two responses were in some agreement, indicating that there may not be a clear benefit to using zWFH over zBMI. Similarly, Aris et al., considered the association of overweight based on zBMI or zWFH and cardiometabolic risk during the first two years of life and found no clear difference using zBMI or zWFH [39].

Both groups of latent trajectories identified using zBMI and zWFH did not show a strong agreement in allocations when contrasted with multivariate zHeight+zWeight. This illustrates additional information the multivariate zHeight+zWeight approach may add to such analyses. The multivariate zHeight+zWeight approach was able to identify abnormal growth with respect to zHeight, zWeight or both while the zWFH and zBMI approaches focus on zWeight with zHeight as a reference regardless of whether zHeight is classified as normal or abnormal. Thus, the use of multivariate zHeight+zWeight has provided a different perspective to our understanding of growth during childhood.

Remarkably, 31% of overweight children under five live in LMICs, while 72% and 59% of wasted and stunted children under five, respectively, live in LMICs. While the prevalence of stunting and wasting in children is decreasing in Southern Africa, the prevalence of obesity is increasing [1]. Fifty percent of subjects within this cohort were identified as stunted at one or more timepoints, in particular the "Sharp Decrease, Sharp Increase to Below Expected" zHeight class. Forty one percent of those allocated to the "Sharp Increase to High" zWeight class were classified as overweight at one or more timepoints. Half of those allocated to the "Sharp Decrease, Sharp Increase to Expected" zWeight class were identified as underweight at one or more timepoints. This illustrates the ability of using identified latent classes as a proxy for those at high risk of stunting, overweight or wasting in a longitudinal setting.

The "Sharp Increase to High" zWeight class identified 93% of RWG children, who are at risk of obesity, and development of non-communicable diseases. Those that experienced RWG make up the majority of children identified within the BMI and WFH "Sharp Increase to High" classes, indicating a higher risk of RWG related adverse effects within these classes. Interestingly the RWG children were allocated to similar classes when considering either zHeight+zWeight or zWeight. Within zHeight+zWeight class (v) 81% of experienced RWG, and hence subjects allocated to this class are at greater risk of RWG associated adverse effects.

Limitations include the use of a single cohort. However, this cohort comprised of almost 1000 children with numerous repeated growth measurements and high cohort retention. Furthermore, this cohort is representative of LMIC child populations with high rates of poverty and infectious diseases, which is a consistently underrepresented population group within latent growth analysis. observations were acquired more frequently during the first year but this is the period of most accelerated growth [43], with growth patterns setting a developmental trajectory for life [25,44]. This may have impacted the shapes of growth curves identified as there was more information to describe these curves during the first year but use of a greater degree of smoothing at early ages may account for this.

To describe the longitudinal growth responses a broken-stick model was used. While more flexible growth models such as a cubic-spline, polynomial or more conventional growth models such as the SITAR could have been used none of these approaches would produce as interpretable results as identified here. The broken-stick model allows direct comparison of changes in growth between respective intervals across latent classes.

Future work will include investigating the association between growth trajectories and childhood or future illness, hence identifying possible areas of intervention to promote optimal growth in children. Here the understanding of how weight and height may change throughout childhood has been broadened. With the identification of these distinct growth trajectories, one may be able to explore whether these specific profiles may serve as an indicator of early onset illness.

## Supporting information

**S1 File. Extended methods.**
(PDF)

**S2 File. Latent Growth Response Analysis Results**
(PDF)

**S1 Fig. Time-line of follow-up visits for subjects within the DCHS. Height, Weight, Head Circumference, BMI and WFH recorded at all visits.**
(TIF)

**S2 Fig.** Diagram illustrating the structure of the multivariate LCMM process when considering distinct latent structures for zHeight and zWeight responses respectively. Additional detail as well as specification of terms can be found within Appendix B in the supplementary materials.
(TIF)

**S3 Fig.** Observed growth measurements over time with a smoothed average indicated with a burgundy line for A) Height, B) Weight and C) BMI.
(TIF)

**S4 Fig.** Standardised growth measurements over time with the average trajectory described through a piecewise-linear spline indicated with a green line for A) zHeight, B) zWeight, C) zWFH and D) zBMI.
(TIF)

**S5 Fig.** Standardised growth measurements over time with the average trajectory described through a piecewise-linear spline indicated with a green line for A) zHeight, B) zWeight, C) zWFH and D) zBMI given additional knots placed at timepoints (0.25,0.75,1,1.5,2,3,4).
(TIF)

**S6 Fig.** Information used to choose the appropriate value for k, the number of latent classes within standardised Height. A) Fit statistics for k = (1:5). Profiles of LCMM Classes identified within standardised Height using a randomly selected 50% of subjects, repeated 10 times for B) k = 3, C) k = 4 and D) k = 5.
(TIF)

**S7 Fig.** Information used to choose the appropriate value for k, the number of latent classes within standardised Weight. A) Fit statistics for k = (1:5). Profiles of LCMM Classes identified within standardised Weight using a randomly selected 50% of subjects, repeated 10 times for B) k = 3, C) k = 4 and D) k = 5.
(TIF)

**S8 Fig.** Information used to choose the appropriate value for k, the number of latent classes within standardised WFH A) Fit statistics for k = (1:5). Profiles of LCMM Classes identified within standardised WFH using a randomly selected 50% of subjects, repeated 10 times for B) k = 3, C) k = 4 and D) k = 5.
(TIF)

**S9 Fig.** Information used to choose the appropriate value for k, the number of latent classes within standardised BMI A) Fit statistics for k = (1:5). Profiles of LCMM Classes identified within standardised BMI using a randomly selected 50% of subjects, repeated 10 times for B) k = 3, C) k = 4 and D) k = 5.
(TIF)

**S10 Fig.** Information used to choose the appropriate value for k, the number of latent classes within zHeight + zWeight. Profiles of LCMM Classes identified within zHeight + zWeight

using a randomly selected 50% of subjects, repeated 10 times for A) k = 3, B) k = 4, C) k = 5 and D) k = 6 illustrated using zHeight.
(TIF)

**S11 Fig.** Information used to choose the appropriate value for k, the number of latent classes within zHeight + zWeight A) Fit statistics for k = (1:6). Profiles of LCMM Classes identified within zHeight + zWeight using a randomly selected 50% of subjects, repeated 10 times for B) k = 3, C) k = 4, D) k = 5 and E) k = 6, illustrated using zWeight.
(TIF)

**S12 Fig.** Information used to choose the appropriate value for k, the number of latent classes within standardised Height, given a piecewise linear spline model specification with knots places at (0.25,0.75,1,1.5,2,3,4). A) Fit statistics for k = (1:5). Profiles of LCMM Classes identified within standardised Height using a randomly selected 50% of subjects, repeated 10 times for B) k = 3, C) k = 4 and D) k = 5.
(TIF)

**S13 Fig.** Information used to choose the appropriate value for k, the number of latent classes within standardised Weight, given a piecewise linear spline model specification with knots places at (0.25,0.75,1,1.5,2,3,4). A) Fit statistics for k = (1:5). Profiles of LCMM Classes identified within standardised Weight using a randomly selected 50% of subjects, repeated 10 times for B) k = 3, C) k = 4 and D) k = 5.
(TIF)

**S14 Fig.** Information used to choose the appropriate value for k, the number of latent classes within standardised WFH, given a piecewise linear spline model specification with knots places at (0.25,0.75,1,1.5,2,3,4). A) Fit statistics for k = (1:5). Profiles of LCMM Classes identified within standardised WFH using a randomly selected 50% of subjects, repeated 10 times for B) k = 3, C) k = 4 and D) k = 5.
(TIF)

**S15 Fig.** Information used to choose the appropriate value for k, the number of latent classes within standardised BMI, given a piecewise linear spline model specification with knots places at (0.25,0.75,1,1.5,2,3,4). A) Fit statistics for k = (1:5). Profiles of LCMM Classes identified within standardised BMI using a randomly selected 50% of subjects, repeated 10 times for B) k = 3, C) k = 4 and D) k = 5.
(TIF)

**S16 Fig.** Information used to choose the appropriate value for k, the number of latent classes within zHeight + zWeight, given a piecewise linear spline model specification with knots places at (0.25,0.75,1,1.5,2,3,4). Profiles of LCMM Classes identified within zHeight + zWeight using a randomly selected 50% of subjects, repeated 10 times for A) k = 3, B) k = 4, C) k = 5 and D) k = 6 illustrated using zHeight.
(TIF)

**S17 Fig.** Information used to choose the appropriate value for k, the number of latent classes within zHeight + zWeight, given a piecewise linear spline model specification with knots places at (0.25,0.75,1,1.5,2,3,4). A) Fit statistics for k = (1:6). Profiles of LCMM Classes identified within zHeight + zWeight using a randomly selected 50% of subjects, repeated 10 times for B) k = 3, C) k = 4, D) k = 5 and E) k = 6, illustrated using zWeight.
(TIF)

**S18 Fig.** Latent Growth Trajectories identified within A) zHeight, B) zWeight, C) zWFH and D) BMI given additional knots placed at timepoints (0.25,0.75,1,1.5,2,3,4).
(TIF)

**S19 Fig.** Latent Growth Trajectories identified within A) zHeight and B) zWeight as identified using the multivariate response of zHeight + zWeight, given additional knots placed at timepoints (0.25,0.75,1,1.5,2,3,4).
(TIF)

**S20 Fig.** Linear link functions illustrating the relationship between zHeight and zWeight and the Latent Process within the multivariate zHeight + zWeight model.
(TIF)

**S21 Fig.** Trajectories of zHeight and zWeight profiles identified given four latent classes when considering a linear or cubic (with three equally spaced knots) spline link function specifying the relationship between the longitudinal outcomes and latent process. Here the effect of link is shown on k = 4; when k = 5 is considered the cubic spline approach identifies a class of outliers (n = 33) which does not meet the criteria for adequate class size, thus leading to k = 4 as the optimal number of classes considered.
(TIF)

**S1 Table.** The mean growth response (with standard deviations) and number of observations at each follow up visit from Birth until 60 months of age.
(XLSX)

**S2 Table** . Summary of Maternal, Child and Socio-Economic Characteristics
(XLSX)

**S3 Table.** Knot locations for the linear mixed effect models fit using a piecewise linear spline to describe the association between the respective growth measures and age.
(XLSX)

**S4 Table.** Fixed Effect Estimates from the Measurement model within the LCMM framework given the zHeight response.
(XLSX)

**S5 Table.** Fixed Effect Estimates from the Measurement model within the LCMM framework given the zWeight response.
(XLSX)

**S6 Table.** Fixed Effect Estimates from the Measurement model within the LCMM framework given the zWFH response.
(XLSX)

**S7 Table.** Fixed Effect Estimates from the Measurement model within the LCMM framework given the zBMI response.
(XLSX)

**S8 Table.** Fixed Effect Estimates from the Measurement model within the LCMM framework given the zHeight response within the zHeight + zWeight model.
(XLSX)

**S9 Table.** Fixed Effect Estimates from the Measurement model within the LCMM framework given the zWeight response within the zHeight + zWeight model.
(XLSX)

**S10 Table.** Comparison of subject allocations to zWFH and multivariate zHeight + zWeight classes. Comparing these allocations when considering trajectories from birth until 5 years of age. (XLSX)

**S11 Table.** Comparison of subject allocations to zBMI and multivariate zHeight + zWeight classes. Comparing these allocations when considering trajectories from birth until 5 years of age. (XLSX)

**S12 Table.** Comparison of subject allocations to zBMI and zWFH classes. Comparing these allocations when considering trajectories from birth until 5 years of age. (XLSX)

**S13 Table.** Proportion of children that experienced rapid weight gain within the first 9 months of life within the three zBMI growth classes. (XLSX)

**S14 Table.** Proportion of children that experienced rapid weight gain within the first 9 months of life within the three zWFH growth classes. (XLSX)

**S15 Table.** Proportion of children that experienced rapid weight gain within the first 9 months of life within the five zHeight + zWeight growth classes. (XLSX)

## Acknowledgments

The authors thank the study and clinical staff at Paarl Hospital, Mbekweni and TC Newman clinics, as well as the CEO of Paarl Hospital, Dr. Kruger, and the Western Cape Health Department for their support of the study. The authors thank the families and children who participated in this study.

## Author contributions

**Conceptualization:** Liz Goddard, Heather J Zar, Francesca Little.

**Data curation:** Noëlle van Biljon, Marilyn T Lake, Maresa Botha.

**Formal analysis:** Noëlle van Biljon, Francesca Little.

**Funding acquisition:** Heather J Zar.

**Investigation:** Noëlle van Biljon, Francesca Little.

**Methodology:** Noëlle van Biljon, Francesca Little.

**Project administration:** Noëlle van Biljon, Liz Goddard, Maresa Botha.

**Supervision:** Liz Goddard, Heather J Zar, Francesca Little.

**Validation:** Noëlle van Biljon.

**Visualization:** Noëlle van Biljon.

**Writing – original draft:** Noëlle van Biljon.

**Writing – review & editing:** Noëlle van Biljon, Marilyn T Lake, Liz Goddard, Heather J Zar, Francesca Little.

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
