## [Decision Letter · Decision Letter 0]

16 Oct 2024

PONE-D-23-38575Latent Classes of Anthropometric Growth in Early Childhood Using Uni- and Multivariate approaches in a South African Birth Cohort.PLOS ONE

Dear Dr. van Biljon,

Thank you for submitting your manuscript to PLOS ONE. After careful consideration, we feel that it has merit but does not fully meet PLOS ONE’s publication criteria as it currently stands. Therefore, we invite you to submit a revised version of the manuscript that addresses the points raised during the review process.

We would like to inquire whether the researchers observed any relationships between rapid weight gain or lack of recovery and maternal BMI or other socioeconomic factors. In Kenya, for example, months of famine occur from February to May, which may intensify challenges for those who do not recover or who experience height recovery. We are also interested in whether they have considered family height targets and head circumference in their analysis.

Please submit your revised manuscript by Nov 30 2024 11:59PM. If you will need more time than this to complete your revisions, please reply to this message or contact the journal office at plosone@plos.org . Please include the following items when submitting your revised manuscript:

We look forward to receiving your revised manuscript.

Kind regards,

Myriam M. Altamirano-Bustamante

Academic Editor

PLOS ONE

Journal requirements: When submitting your revision, we need you to address these additional requirements. 1. Please ensure that your manuscript meets PLOS ONE's style requirements, including those for file naming. The PLOS ONE style templates can be found at https://journals.plos.org/plosone/s/file?id=wjVg/PLOSOne_formatting_sample_main_body.pdf and https://journals.plos.org/plosone/s/file?id=ba62/PLOSOne_formatting_sample_title_authors_affiliations.pdf 2. Thank you for stating the following financial disclosure:  [This study was funded by Bill and Melinda Gates Foundation (OPP 1017641; OPP1017579), the NIH, USA (U54HG009824, U01AI110466], South African Medical Research Council, and the University of Cape Town (UCT). NVB was supported through a fellowship to UCT from the Developing Emerging African Leaders (DEAL) programme by The Carnegie Corporation of New York and a Departmental Fellowship from the Department of Statistical Sciences, UCT.].  Please state what role the funders took in the study.  If the funders had no role, please state: ""The funders had no role in study design, data collection and analysis, decision to publish, or preparation of the manuscript."" If this statement is not correct you must amend it as needed. Please include this amended Role of Funder statement in your cover letter; we will change the online submission form on your behalf. 3. We note that you have indicated that there are restrictions to data sharing for this study. For studies involving human research participant data or other sensitive data, we encourage authors to share de-identified or anonymized data. However, when data cannot be publicly shared for ethical reasons, we allow authors to make their data sets available upon request. For information on unacceptable data access restrictions, please see http://journals.plos.org/plosone/s/data-availability#loc-unacceptable-data-access-restrictions.  Before we proceed with your manuscript, please address the following prompts: a) If there are ethical or legal restrictions on sharing a de-identified data set, please explain them in detail (e.g., data contain potentially identifying or sensitive patient information, data are owned by a third-party organization, etc.) and who has imposed them (e.g., a Research Ethics Committee or Institutional Review Board, etc.). Please also provide contact information for a data access committee, ethics committee, or other institutional body to which data requests may be sent. b) If there are no restrictions, please upload the minimal anonymized data set necessary to replicate your study findings to a stable, public repository and provide us with the relevant URLs, DOIs, or accession numbers. Please see http://www.bmj.com/content/340/bmj.c181.long for guidelines on how to de-identify and prepare clinical data for publication. For a list of recommended repositories, please see https://journals.plos.org/plosone/s/recommended-repositories. You also have the option of uploading the data as Supporting Information files, but we would recommend depositing data directly to a data repository if possible. Please update your Data Availability statement in the submission form accordingly. 4. Your ethics statement should only appear in the Methods section of your manuscript. If your ethics statement is written in any section besides the Methods, please move it to the Methods section and delete it from any other section. Please ensure that your ethics statement is included in your manuscript, as the ethics statement entered into the online submission form will not be published alongside your manuscript.  5. Please include captions for your Supporting Information files at the end of your manuscript, and update any in-text citations to match accordingly. Please see our Supporting Information guidelines for more information: http://journals.plos.org/plosone/s/supporting-information. 

Reviewers' comments:

Reviewer's Responses to Questions

**Comments to the Author**

1. Is the manuscript technically sound, and do the data support the conclusions?

Reviewer #1: Yes

Reviewer #2: Yes

2. Has the statistical analysis been performed appropriately and rigorously? 

Reviewer #1: Yes

Reviewer #2: Yes

3. Have the authors made all data underlying the findings in their manuscript fully available?

Reviewer #1: Yes

Reviewer #2: Yes

4. Is the manuscript presented in an intelligible fashion and written in standard English?

Reviewer #1: Yes

Reviewer #2: Yes

5. Review Comments to the Author

Reviewer #1: This study aims to identify growth patterns in children from birth to 5 years of age in a South African birth cohort using a latent class mixed modelling. Much information was given in the supplemental materials, but their results were easy to be followed. The patterns of abnormal child growth, such as stunting, underweight and overweight, have been a focus of research and assessed for various health outcomes in later life. This study provided a useful tool on the early childhood growth patterns and may serve for a potential intervention to prevent the diseases in adulthood. This manuscript can be published with some minor modifications.

1. In abstract (results), “…strong association was found between various growth classes and abnormal growth features….” The word “strong” need to be explained with values in the abstract.

2. In the manuscript, the words of “Height” and “height,” and “Weight” and “weight” were mixed. They need to be uniformed.

3. In methods, it is stated that “Pregnant women in a poor peri-urban area in South Africa, were enrolled during their second trimester of pregnancy, from March 2012 – March 2015.” It should be explained more on the study population and selection procedures, e.g., target community population, expected annual number of pregnancies, number of identified pregnancies by the study, selection criteria, urine pregnancy test done by the study.

4. The standardized anthropometric values depend on the accuracy of gestational age at birth. How this study assured the accuracy of gestational age at birth. This might be explained in a previous publication of this cohort, but this point is critical for this modelling study, it is important to explain it, e.g., the last menstruation period is based on mother’s recall, or the study used ultrasound-based gestational age at enrollment?

5. Page 6, Line 2, “Measures within 0.5cm and 0.1kgs…” should be corrected as “Measures within 0.5cm and 0.1kg….”

6. It might be more interesting for international readers if other ethnicities studies can be discussed on the early childhood growth patterns. As this study demonstrated, there has been studies reporting that sub-Saharan African children were mostly born in normal size, then deteriorate towards 2-3 years of age. On the other hand, South Asian children tend to be born small and track various pattern during the childhood. It is important to discuss these patterns can be discussed with the internal (biological) and external (environmental) factors.

7. There is no discussion on the mode of delivery. This relates to distinguished characteristics of newborn gut environment as reported in many publications. Moreover, cesarian section has been rapidly increased in LMIC.

Reviewer #2: The objective of this study is very specific: to demonstrate growth patterns in children, focusing primarily on weight expressed in terms of weight for height. By utilizing English and New Zealand growth charts, the study incorporates a definition of rapid weight gain, defined as greater than 0.67 standard deviations (SD).

One of the key questions addressed is how to identify children who experience rapid or slow weight gain. The following points are critical for this analysis:

The ideal timing for obtaining measurements.

The cutoff point for identifying growth patterns. The study employs a method similar to that used for English children, utilizing 9 centiles, which corresponds to 0.67 SD between each centile. This approach appears to be more effective than the WHO standards, which classify overweight as weight for height greater than 2 SD and obesity as greater than 3 SD.

Additionally, we would like to inquire whether the researchers observed any relationships between rapid weight gain or lack of recovery and maternal BMI or other socioeconomic factors. In Kenya, for example, months of famine occur from February to May, which may intensify challenges for those who do not recover or who experience height recovery. We are also interested in whether they have considered family height targets and head circumference in their analysis.

6. PLOS authors have the option to publish the peer review history of their article (what does this mean? ). If published, this will include your full peer review and any attached files.

**Do you want your identity to be public for this peer review?** For information about this choice, including consent withdrawal, please see our Privacy Policy .

Reviewer #1: No

Reviewer #2: No

---

## [Author Response · Author response to Decision Letter 0]

17 Jan 2025

Reviewer #1: This study aims to identify growth patterns in children from birth to 5 years of age in a South African birth cohort using a latent class mixed modelling. Much information was given in the supplemental materials, but their results were easy to be followed. The patterns of abnormal child growth, such as stunting, underweight and overweight, have been a focus of research and assessed for various health outcomes in later life. This study provided a useful tool on the early childhood growth patterns and may serve for a potential intervention to prevent the diseases in adulthood. This manuscript can be published with some minor modifications.

1. In abstract (results), “…strong association was found between various growth classes and abnormal growth features….” The word “strong” need to be explained with values in the abstract.

To clarify this, the following was added to the abstract:

Much greater or lesser than expected proportions (an increase by 25% in some cases), when compared to the proportion of abnormal growth features across the entire cohort.

2. In the manuscript, the words of “Height” and “height,” and “Weight” and “weight” were mixed. They need to be uniformed.

Thank you for this observation. Within the manuscript all “Height” have been changed to “height”, and all “Weight” have been changed to “weight”.

3. In methods, it is stated that “Pregnant women in a poor peri-urban area in South Africa, were enrolled during their second trimester of pregnancy, from March 2012 – March 2015.” It should be explained more on the study population and selection procedures, e.g., target community population, expected annual number of pregnancies, number of identified pregnancies by the study, selection criteria, urine pregnancy test done by the study.

Thank you for this comment, to elaborate details with respect to recruitment we have prepared the following:

Mothers were recruited at an antenatal clinic, and hence pregnancy was already established by the primary healthcare providers. Therefore, no pregnancy test were performed by study staff. Mothers were enrolled at 20-28 weeks as this is when the antenatal scans are performed.

The following was added to the methods section:

Mothers were recruited at an antenatal clinic, and all who were over the age of 18 and intended to reside in the Paarl area for at least one year were eligible for this study.

4. The standardized anthropometric values depend on the accuracy of gestational age at birth. How this study assured the accuracy of gestational age at birth. This might be explained in a previous publication of this cohort, but this point is critical for this modelling study, it is important to explain it, e.g., the last menstruation period is based on mother’s recall, or the study used ultrasound-based gestational age at enrollment?

Thank you for this important comment, to respond to this the following was added to the methods section:

Gestational age was calculated using second trimester ultrasound. In the event that this was unavailable, last menstrual period was used to calculate gestational age at the time of enrolment.

5. Page 6, Line 2, “Measures within 0.5cm and 0.1kgs…” should be corrected as “Measures within 0.5cm and 0.1kg….”

This change has been made.

6. It might be more interesting for international readers if other ethnicities studies can be discussed on the early childhood growth patterns. As this study demonstrated, there has been studies reporting that sub-Saharan African children were mostly born in normal size, then deteriorate towards 2-3 years of age. On the other hand, South Asian children tend to be born small and track various pattern during the childhood. It is important to discuss these patterns can be discussed with the internal (biological) and external (environmental) factors.

Thank you for this point, to comment on this we have included the following to the discussion:

Stein et al., (2010) compared the patterns of height in early childhood (birth to age 5) in five birth cohorts from distinct populations around the globe (Brazil, Guatemala, India, Philippines and South Africa) (18). Considering the same cohorts, Poveda Rey et al., (2021) looked at weight throughout early childhood (19). The Brazilian cohort mostly followed expected standardised height and expected standardised weight within the first five years of life. While the Guatemalan, South African and Indian cohorts were born smaller (with respect to length) than the expected; the South African cohort recovered by the age of five while the Guatemalan and Indian cohorts did not. The Guatemalan, Filipino and South African cohorts were born weighing slightly less than expected. The South African cohort did not recover, nor did they show further reduction in expected weight. However, the Guatemalan and Filipino cohorts did show further reduction in standardised weight scores, to a greater extent for the Filipino cohort. The Filipino cohort was born with an expected height but decelerated to much smaller than expected by one year of age. The Indian cohort was born with the smallest standardised weight which did not improve through early childhood. Interestingly, all cohorts showed a decreased standardised height at two years compared to birth; while some were able to recover from this deceleration (Brazil, South Africa), other cohorts did not (Guatemala, India, Philippines).

7. There is no discussion on the mode of delivery. This relates to distinguished characteristics of newborn gut environment as reported in many publications. Moreover, cesarian section has been rapidly increased in LMIC.

Thank you for this comment. We have added the following to the methods section:

The most common mode of delivery was natural birth, while 20.12% of the children within the study were delivered by caesarean-section.

Reviewer #2: The objective of this study is very specific: to demonstrate growth patterns in children, focusing primarily on weight expressed in terms of weight for height. By utilizing English and New Zealand growth charts, the study incorporates a definition of rapid weight gain, defined as greater than 0.67 standard deviations (SD).

One of the key questions addressed is how to identify children who experience rapid or slow weight gain. The following points are critical for this analysis:

1. The ideal timing for obtaining measurements.

To clarify this, the following was added into the methods section:

The most rapid period of growth through childhood is during infancy, specifically throughout the first 2 years, hence measurements were captured more frequently through this period.

To explain this in detail, we have prepared the following, however this has not been added into the manuscript:

After birth, there is an initial period of accelerated growth which then slows down by one year of age. Considering weight, children double their birth weight around 5-6 months of age and triple their weight it at one year of age. Between 1-2 years children then gain 2.5 kg on average. Considering height, from birth to 6 months infants grow on average 2.5 cm a month which slows to 1.25 cm a month from 7 to 12 months and then to 10 to 12 cm between 1 and 2 years.

2. The cutoff point for identifying growth patterns. The study employs a method similar to that used for English children, utilizing 9 centiles, which corresponds to 0.67 SD between each centile. This approach appears to be more effective than the WHO standards, which classify overweight as weight for height greater than 2 SD and obesity as greater than 3 SD.

Here there may be a misunderstanding of our approach used, to clarify this we respond with:

To prepare the observed height, weight and BMI data for the longitudinal modelling and latent class analysis process the WHO approach of standardisation was used. This approach compares observed data to expected height/ weight/ bmi reference ranges given child age and gender to allocate a zScore. This zScore is close to 0 if a child has a growth measurement close to the expected value for their specific age and gender, or a larger positive or negative value if the child is smaller or larger than expected respectively. Once these zScores are calculated no cut-offs are employed within the longitudinal modelling and latent class identification process. If the reviewer perhaps meant to question the cutoff used with respect to RWG, as mentioned in the methods, RWG during the first nine months was identified. However, when identifying the proportion of children belonging to abnormal growth profiles (stunted, obese or overweight) within the latent growth profiles then the WHO standards were indeed used.

3. Additionally, we would like to inquire whether the researchers observed any relationships between rapid weight gain or lack of recovery and maternal BMI or other socioeconomic factors. In Kenya, for example, months of famine occur from February to May, which may intensify challenges for those who do not recover or who experience height recovery. We are also interested in whether they have considered family height targets and head circumference in their analysis.

To answer this briefly, yes. However, this is not in the scope of this analysis. We are currently working on a separate paper that explores the relationship between latent growth profiles and determinants of such growth which includes socioeconomic factors and measures of maternal BMI as well as many other variables. We chose to focus on these as separate analyses to allow a thorough and detailed exploration of both class identification and predictor analysis. We have not looked at family height targets and head circumference as these measures still require a large amount of data cleaning before they ready to be used, and once again this is not in the scope of this analysis as we have focused on Height, Weight and composite measures of these two variables.

---

## [Decision Letter · Decision Letter 1]

30 Jan 2025

Latent Classes of Anthropometric Growth in Early Childhood Using Uni- and Multivariate approaches in a South African Birth Cohort.

PONE-D-23-38575R1

Dear Dr. Noëlle van Biljon

We’re pleased to inform you that your manuscript has been judged scientifically suitable for publication and will be formally accepted for publication once it meets all outstanding technical requirements.

Kind regards,

Myriam M. Altamirano-Bustamante

Academic Editor

PLOS ONE

Additional Editor Comments (optional):

Reviewers' comments:

Reviewer's Responses to Questions

**Comments to the Author**

1. If the authors have adequately addressed your comments raised in a previous round of review and you feel that this manuscript is now acceptable for publication, you may indicate that here to bypass the “Comments to the Author” section, enter your conflict of interest statement in the “Confidential to Editor” section, and submit your "Accept" recommendation.

Reviewer #1: All comments have been addressed

2. Is the manuscript technically sound, and do the data support the conclusions?

Reviewer #1: Yes

3. Has the statistical analysis been performed appropriately and rigorously? 

Reviewer #1: Yes

4. Have the authors made all data underlying the findings in their manuscript fully available?

Reviewer #1: No

5. Is the manuscript presented in an intelligible fashion and written in standard English?

Reviewer #1: Yes

6. Review Comments to the Author

Reviewer #1: (No Response)

7. PLOS authors have the option to publish the peer review history of their article (what does this mean? ). If published, this will include your full peer review and any attached files.

**Do you want your identity to be public for this peer review?** For information about this choice, including consent withdrawal, please see our Privacy Policy .

Reviewer #1: No

---

## [Editor Report · Acceptance letter]

PONE-D-23-38575R1

PLOS ONE

Dear Dr. van Biljon,

I'm pleased to inform you that your manuscript has been deemed suitable for publication in PLOS ONE. Congratulations! Your manuscript is now being handed over to our production team.

Kind regards,

on behalf of

Dr. Myriam M. Altamirano-Bustamante

Academic Editor

PLOS ONE